# Environmental Transmission of Symbionts in the Mangrove Crabs *Aratus pisonii* and *Minuca rapax*: Acquisition of the Bacterial Community through Larval Development to Juvenile Stage

**DOI:** 10.3390/microorganisms12040652

**Published:** 2024-03-25

**Authors:** Naëma Schanendra Béziat, Sébastien Duperron, Olivier Gros

**Affiliations:** 1Institut de Systématique, Evolution, Biodiversité (ISYEB), Muséum National d’Histoire Naturelle, CNRS, Sorbonne Université, EPHE, Université des Antilles, Campus de Fouillole, 97110 Pointe-à-Pitre, France; olivier.gros@univ-antilles.fr; 2Caribaea Initiative, Université des Antilles, 97110 Pointe-à-Pitre, France; 3Molécules de Communication et Adaptation des Microorganismes (MCAM), Muséum National d’Histoire Naturelle, UMR 7245, CNRS, 57 Rue Cuvier (CP54), 75005 Paris, France; sebastien.duperron@mnhn.fr; 4C3MAG, UFR des Sciences Exactes et Naturelles, Université des Antilles, 97110 Pointe-à-Pitre, France

**Keywords:** environmental transmission, microbiology, symbiosis, crustaceans, marine biology, bacteria, ectosymbionts

## Abstract

*Aratus pisonii* and *Minuca rapax* are two brachyuran crabs living with bacterial ectosymbionts located on gill lamellae. One previous study has shown that several rod-shaped bacterial morphotypes are present and the community is dominated by Alphaproteobacteria and Bacteroidota. This study aims to identify the mode of transmission of the symbionts to the new host generations and to identify the bacterial community colonizing the gills of juveniles. We tested for the presence of bacteria using PCR with universal primers targeting the 16S rRNA encoding gene from gonads, eggs, and different larval stages either obtained in laboratory conditions or from the field. The presence of bacteria on juvenile gills was also characterized by scanning electron microscopy, and subsequently identified by metabarcoding analysis. Gonads, eggs, and larvae were negative to PCR tests, suggesting that bacteria are not present at these stages in significant densities. On the other hand, juveniles of both species display three rod-shaped bacterial morphotypes on gill lamellae, and sequencing revealed that the community is dominated by Bacteroidota and Alphaproteobacteria on *A. pisonii* juveniles, and by Alphaprotobacteria, Bacteroidota, and Acidimicrobia on *M. rapax* juveniles. Despite the fact that juveniles of both species co-occur in the same biotope, no shared bacterial phylotype was identified. However, some of the most abundant bacteria present in adults are also present in juveniles of the same species, suggesting that juvenile-associated communities resemble those of adults. Because some of these bacteria were also found in crab burrow water, we hypothesize that the bacterial community is established gradually during the life of the crab starting from the megalopa stage and involves epibiosis-competent bacteria that occur in the environment.

## 1. Introduction

Most eukaryotes live in symbiotic association with prokaryotic partners. These associations are particularly well documented in animals living in various biotopes from terrestrial to aquatic environments [1]. De Bary (1879) proposed the first definition of symbiosis as the “living together of dissimilarly named organisms”. This definition encompasses all types of relationships, and describes symbiosis in the broadest sense [2]. Many criteria must be considered to characterize a relationship between a eukaryotic host and its symbionts, including the specificity and duration of the association, its obligatory or facultative character, the localization of the symbionts, the metabolic exchanges, and also their transmission mode to the next host generations [3]. From an evolutionary and survival perspective, it is important for hosts to ensure efficient transmission of their symbionts [4]. Two modes of transmission have been distinguished: vertical and horizontal, the latter including the case of environmental transmission. Vertical transmission occurs via the gametes [5]. Most often the symbionts are transmitted by the female. Symbionts can be located on the egg’s surface, as described in the insect *Cavelerius saccharivorus* [6], or within the oocyte/egg, as described in whiteflies colonized by *Rickettsia* [7] or in the clam *Solemya reidi* colonized by sulfur-oxidizing bacteria [8]. Vertical transmission leads to a high specificity and fidelity [9] and, in some species, results in co-speciation events between hosts and symbionts [10]. Horizontal transmission, on the other hand, involves an aposymbiotic phase in larvae or juveniles followed by the symbiotic phase for the rest of the host’s life, usually after metamorphosis. Horizontal transmission can be inter- or intra- specific as observed for *Wolbachia* spp. Bacteria and diverse hosts [11,12], or can occur through contamination by environmental bacteria as for the squid *Euprymna scolopes* and its associated light-producing symbiont *Vibrio fischeri* [13] or in lucinid bivalves [14]. In this latter case, a stock of symbiosis-competent bacteria naturally present in the host’s environment has to be captured by the aposymbiotic host [4,14,15,16].

In marine crustaceans, symbioses have been less documented than in other groups such as mollusks [10,17,18]. When present, symbionts tend to be ectosymbiotic. They can be located on the cuticle as in shrimps of the genus *Rimicaris* [19] and in the crab *Kiwa hirsuta* [20], or on gills in the lobster *Homarus gammarus* [21]. The mode of symbiont acquisition is poorly characterized in crustaceans. Yet, bacterial symbionts present on the carapace of *Kiwa* crabs are, for example, present in the environment [22,23], suggesting environmental transmission for those species.

Despite limited research effort, some species of coastal crabs from Africa and Asia are reported to be associated with bacteria located on their gills, including the freshwater species *Eriocheir sinensis,* mainly colonized by Actinobacteria, Proteobacteria, Bacteroidota, and Firmicutes [24,25], but no data are available concerning the transmission mode of the symbionts in the species concerned. In a recent study focusing on two Caribbean mangrove brachyuran decapods [26], we have shown that adult *Aratus pisonii* (arboreal mangrove crab) and *Minuca rapax* (living on mangrove mud) are associated with specific lineages of bacteria, dominated by Bacteroidota, Actinobacteria, and Alphaproteobacteria. While similar at the phylum level, bacterial communities differed at the 16S rRNA level between the two species. These two species were chosen as models because they are abundant in Caribbean mangrove and easy to sample, and because *M. rapax* is an important species in the mangrove, feeding on leaf litter and other organic matter, thus actively recycling nutrients and oxygenating sediment through its burrowing activity. It was also the first report of bacterial symbionts colonizing the gills of these two species in Guadeloupe. This new study follows on from the previous one, this time investigating other life stages of these two species.

The aim of this study is to investigate how the symbiotic bacterial populations are transmitted to the next generations in *Aratus pisonii* and *Minuca rapax*. Juveniles of both species share the same environment, while adults use different habitats (trees [27] and mud [28], respectively). For *A. pisonii*, larval development lasts over 19 days [29], and sexual maturity is reached when the carapace measures over 10 mm (Carapace length) [30]. For *M. rapax*, larval development lasts over 13 days [31]. Sexual maturity is also reached when the carapace measures over 10 mm (CL), which also depends on the geographic area studied [32].

We thus investigate if the environment has more impact on the community composition than the host species. To this purpose, the presence of bacteria was investigated in gonads, eggs, larvae, and juveniles using PCR and electron microscopy analyses. Besides those collected in the field, larvae were also obtained in the laboratory and reared in controlled conditions. The composition of gill-associated bacterial communities in juveniles was analyzed and compared to that of adult specimens as well as environmental samples (mangrove tree bark and burrow’s water).

## 2. Materials and Methods

### 2.1. Sampling Site and Crab Collection

During this study, 6 juveniles and 15 females were collected for both species. Concerning DNA analysis, 2 juvenile specimens of each species were used. For scanning electron microscope analysis, we observed 4 juveniles per species.

Ovigerous females and juveniles of the crabs *Aratus pisonii* and *Minuca rapax* were collected manually from the mangrove located in the Manche à Eau lagoon in Guadeloupe (Lesser Antilles, collection site: N16°27′459″/W61°55′722″ (Figure 1). *Aratus pisonii* adults were collected directly from *Rhizophora mangle* trees while juveniles were collected only from the sediment. Adults and juveniles of *Minuca rapax* were collected from the sediment from the same area. During collection, two *A. pisonii* adults were collected just after molt. *Aratus pisonii* ovigerous females were collected between January and March 2019, while *Minuca rapax* ovigerous females were collected in July 2020. For each egg-laying attempt, about 10 females of each species were transferred to the laboratory for natural spawning and to obtain larval culture.

The gills from adult individuals were collected using tweezers after a cold anesthesia followed by the opening of the carapace as previously described [26]. We also collected gonads and egg grape from two different females (Appendix A).

Swimming larvae (zoea stages) were captured from the same location in February 2021 with a plankton net (200 µm mesh size) during a transect across the entire lagoon near the mangrove fringe with a small boat operated at 2 knots.

Water samples from burrows of *M. rapax* were collected in duplicate at low tide using a 50 mL syringe. Six milliliters per burrow was centrifuged at 10,000× *g* for 3 min. The pellet obtained was pooled for DNA extraction. Two burrows were analyzed using metabarcoding analysis. Surface of *Rhizophora mangle* bark was collected by scraping using sterile tweezers. Several roots and branches were scraped to produce one sample. Upon recovery, four samples were frozen. Two were used for DNA analysis.

### 2.2. Larval Culture

*Aratus pisonii* and *Minuca rapax* ovigerous females were kept in the laboratory in a dry container, with access to seawater for egg laying. During the night, the crabs came out of their container to release eggs into the water. After hatching (usually early in the morning the day after capture), all the larvae were collected with a 50 mL syringe and transferred into a cylindroconic batch containing 4 L of 1.2 µm-filtered seawater (salinity: 34.6 g.L^−1^). Sea water was gently oxygenated in depth and renewed every two days to avoid bacterial development. The larvae were fed with *Artemia nauplii* (3 individuals per mL) every two days.

### 2.3. DNA Extraction and Polymerase Chain Reaction Analyses

DNA extraction and PCR analyses were performed on gonads from two adult females, and the eggs, larvae, and gills of two juveniles per species. Regarding the larvae either collected from the field or obtained in laboratory, about ten zoea were extracted, and one megalopa. We call “larvae” the set of zoea and megalopa, although extractions of the zoea and the megalopa were performed separately. A cold anesthesia of all individuals investigated was performed before dissection. Environmental DNA was also extracted from four samples of the scraped surface of *R. mangle* and from burrow water samples. For the latter, twenty milliliters of water were centrifuged at 5000× *g* for 5 min and DNA was extracted from the obtained pellet.

DNA extractions were performed using the DNeasy Blood & Tissue Kit (QIAGEN, Hilden, Germany) following manufacturer’s instructions. DNA from gills of *A. pisonii* and *M. rapax* adults was used as a positive control, while ultra-pure water (H_2_O) was used as a negative control. We performed a positive control by amplifying the host 18S rRNA-encoding gene PCR with universal primers for all samples [33] First, we performed a first screen for the presence of bacteria in various samples by PCR using universal primer set (8F-1492R) [34,35]. Specific primer sets (8F-CFB 563R; 8F-ROS 537R) targeting the bacterial groups (Cytophaga, Flavobacterium, Bacteroides, and *Roseobacter*) present in adults of *A. pisonii* and *M. rapax* [26] were also used on these same samples [34,36,37,38] (Table 1, Appendix A). Because different sets of primers could have different “sensitivities” to detect bacterial DNA, 3 different primer sets were used in case DNA was present in small quantities. Standard conditions were used for PCR, with initial denaturation at 94 °C for 4 min, 30 cycles at 94 °C for 1 min, at 52 °C for 45 s, and 72 °C for 1.5 min, and a final elongation for 7 min at 72 °C. PCR-positive samples were then used for metabarcoding analyses (Table 1, see next paragraph, Appendix A).

### 2.4. Composition of Gill-Associated Bacterial Communities Based on 16S rRNA-Encoding Gene Sequencing and Phylogenetic Reconstruction

Metabarcoding analyses were performed on juveniles’ gills and environmental samples (burrow waters and *R. mangle* cortex). A~400 bp fragment of the rRNA-encoding gene corresponding to the V4-V5 variable region of *Escherichia coli* was amplified using primers 515F (5′-GTGYCAGCMGCCGCGGTAA-3′) and 926R (5′-CCGYCAATTYMTTTRAGTTT-3′) [39] and sequenced on an Illumina MiSeq platform (2 X 300 bp, paired-end sequencing, Genoscreen, France). Company-provided mock communities of known composition were used as an internal control for the whole sequencing process. Resulting sequencing dataset was deposited in Sequence Read Archive under project PRJNA958136 (samples SAMN34279703-10).

Sequence analyses were performed using QIIME2 [40]. Amplicon Sequence Variants (ASVs [41]) were identified with DADA2 using default parameters, i.e., a maximal probability for indels of 0.01 and mean read error rate of 0.5% for normalization. DADA2 was also used to identify and remove chimeric sequences. Taxonomic affiliations were affiliated using the SILVA 138-99 database.

ASVs obtained were compared with ASVs recently obtained from the gills of adult specimens [26]. These were included along with a dataset of reference sequences and sequences of close relatives (according to BLAST made on NCBI website). Sequences were aligned using ClustalW implemented in MEGA 11 [42,43], alignment was trimmed to ASV region, and the phylogenetic tree was reconstructed with the same software (MEGA 11) [43]. The tree was inferred based on 341 positions using the Maximum Likelihood method and a General Time-Reversible model with a Gamma distribution of evolutionary rates (5 categories and invariants).

### 2.5. Ultrastructural Analysis

For scanning electron microscopy (SEM) observations, samples of eggs from ovigerous females collected in the field, gills of juveniles of both species, and gills of the post-molt specimens were used. Juvenile gills used for ultrastructural analysis and metabarcoding analysis were from the same individual.

Tissues were fixed at 4 °C in 2.5% glutaraldehyde in 0.8× PBS buffer (pH 7.2) for at least 2 h. They were then dehydrated in series of acetone solutions of increasing concentration (30°, 50°, 70°, 90°, and 3 times at 100°), critical-point-dried with CO_2_, and sputter-coated with gold before observation with a FEI Quanta 250 electron microscope at 20 kV (Illkirch, France).

## 3. Results

### 3.1. Analyses of Gonads and Eggs

The universal and specific PCR amplifications of the 16S rDNA gene from gonads and eggs of *A. pisonii* and *M. rapax* using the three primer sets were negative despite positive results on controls used (Table 1).

Examination using SEM failed to detect bacterial symbionts on the surface of the eggs of *A. pisonii* (Figure 2A). These results suggest a very rare occurrence of bacteria on eggs and gonads, below the detection limit of the methods used.

### 3.2. Analyses on Wild and Lab-Reared Larval Stages

The larvae of *A. pisonii* collected in the field had a carapace length of 412.8 ± 62.20 µm (n = 20). The lab-reared larvae of *A. pisonii* had a carapace length of 332.06 ± 54.46 µm (n = 30) and *M. rapax* lab-reared larvae had a carapace length of 323.80 ± 12.14 µm (n = 21).

The zoea 1 larval stages obtained in the laboratory were tested just after hatching for both species (Figure 3). PCR amplifications with universal and group-specific primer sets were negative in both *A. pisonii* and *M. rapax* (Table 1). The same result was obtained for *A. pisonii* field-sampled larvae, whatever the stage. Neither zoea stages 1 and 3 nor megalopa tested separately yielded a band in PCR (Table 1).

### 3.3. Bacterial Communities Associated with Wild Juveniles

The measured juvenile specimens collected in the field had a carapace length (CL) of 5.6 ± 1.3 mm for *A. pisonii* and 6.1 ± 1.6 mm for *M. rapax*.

Rod-shaped bacteria of different sizes and shapes were present on gill lamellae of juveniles of *A. pisonii* and *M. rapax* (Figure 2B–D). Three different morphotypes were observed on *A. pisonii*: a short and thin morphotype, a short and large morphotype, and a long morphotype. *M. rapax* also yielded three morphotypes: a big morphotype, a thin morphotype, and a short morphotype. In comparison, the post-molt adult of *A. pisonii* displayed no bacteria on the surface of its gills (Figure 2C). Positive PCR amplifications with all three primer sets were obtained from the same *M. rapax* and *A. pisonii* juveniles observed using SEM and collected from the sediment (Table 1).

An analysis of community compositions based on 16S rRNA V4–V5 sequencing in the two *Aratus pisonii* juveniles yielded 40,699 and 37,207 quality-filtered reads clustering into 65 and 56 ASVs, respectively (combined samples: 88 ASVs, Figure 4). Alphaproteobacteria represented 56 and 58%, and Bacteroidota represented 42 and 40% of the reads (Figure 5). Seven ASVs are abundant (each representing above 1% of the reads) in *A. pisonii*. ARA-juv-1 is the most abundant, with 34% and 40% of the reads in the two specimens; it belongs to the Alphaproteobacteria group and is identical (100% sequence similarity) to an uncultured bacterium found on gills of the mangrove crab *Perisesarma guttatum* (KT944808, Figure 6). ARA-juv-2 represents 13% and 7% of the two specimens, it also belongs to Alphaproteobacteria, and is 99% similar to an uncultured bacterium found in sediment in Chile (EF632822). ARA-juv-3 to 7 all belong to the Bacteroidota group. Some are similar to environmental sequences. ARA-juv-3 represents 12% of the reads and is 94% similar to an uncultured bacterium found in the gastropod *Triona metcalfi* (OL862969) and ARA-juv-4 represents 7% and 1% of the reads on the two specimens and is 97% similar to an uncultured organism from environmental samples (JN522769). ARA-juv-5 represents 6% and 2% of the reads and is 90% similar to an uncultured bacterium found in the sponge *Tethya californiana* (EU290362). ARA-juv-6 represents 3% and 6% of the reads and is 89% similar to an uncultured bacterium found in environmental samples (KM456175). ARA-juv-7 represents 3% and 8% of the reads and is 90% similar to an uncultured bacterium found in environmental samples.

The two *Minuca rapax* juveniles yielded 14.777 and 15.513 quality-filtered reads, representing 17 ASVs for both samples (combined samples: 32 ASVs, Figure 4). Alphaproteobacteria represented 84 and 88% of the reads in the two juveniles examined, respectively (Figure 5). Bacteroidota were below 15% while Actinobacteria represented around 1%. Four ASVs are abundant in *M. rapax* juveniles. By far the most abundant ASV is MIN-juv-1, representing 86% of the community in the first individual and 79% of the community in the second one. It corresponds to an Alphaproteobacteria and is 100% identical to an uncultured bacterium found in gills of the crab *Uca urvillei* (KT944883, Figure 6). MIN-juv-2 represents 3% of the reads and is present only on one specimen; it belongs to the Bacteroidota group, and it is 94% similar to an uncultured bacterium found on a macroalgal surface (GU451338). MIN-juv-3 represents 1% and 13% of the reads, it belongs to the Bacteroidota group and is 93% similar to an uncultured bacterium found in *Uca urvillei* gills (KT944910). MIN-juv-4 represents 1% of the reads (present only in one sample). It is also a Bacteroidota similar at 92% to an uncultured bacterium found in the gills of the crab *Perisesarma guttatum* (KT944898). A supplementary table specifies the taxonomic group obtained for each main ASV of *A. pisonii* and *M. rapax* (Appendix A).

No ASV is shared between the juveniles of *A. pisonii* and *M. rapax* (Figure 4).

### 3.4. Comparison with Communities from Tree Bark, Burrow Water, and Adult Crabs

Two samples of *Rhizophora mangle* yielded 30,044 and 26,579 quality-filtered reads clustered into 87 and 92 ASVs, respectively. The samples were dominated by Rhodothermia (32.8% and 34.4% of reads) and Actinobacteria (39.3% and 28.6%, Figure 5). No ASVs were shared between *Minuca rapax* or *Aratus pisonii* juveniles and *Rhizophora mangle* samples (Figure 4).

In the two burrow water samples analyzed, 19,186 and 15,704 quality-filtered reads were obtained, clustering into 122 and 402 ASVs, respectively. Alphaproteobacteria was the most abundant phylum in one of the samples (46.5% of reads), while the others were dominated by Gammaproteobacteria (76.4% of reads). A total of 15 ASVs were shared between burrow waters and *A. pisonii juveniles* (Figure 4). Two of the seven abundant ASVs found in *A. pisonii* juveniles, namely ARA-juv-1 and ARA-juv-2, were 99% similar to two ASVs found in burrow waters. ARA-juv-1 was the most abundant ASV on *A. pisonii* juvenile and represents 10% of the reads from burrow waters (5 nt difference over 372 nt). ARA-juv-2 was similar to one ASV that represents 1% of the reads from burrow waters (2 nt difference over 372 nt). The other ASVs shared between *A. pisonii* and burrow waters are rare in the latter, representing between 0.01 and 0.4% of the reads for *A. pisonii* samples and 0.05% and 8.7% of the reads for burrow waters samples. These ASVs are present only in one sample of burrow waters.

A comparison was then made then with the five main ASVs previously found in adult *A. pisonii* gills, namely, ASV Ara-1, -2, -3, -4, and -5 [26]. ASVs ARA-juv-1 and ARA-Juv-5 are 100% identical to ASV Ara-1 and ASV Ara-3 obtained in adults, respectively, as confirmed in the phylogenetic reconstruction (Figure 6). ARA-juv-1/ASV Ara-1 is the most abundant ASV in both adult and juvenile specimens (34% and 40% of the reads for juvenile samples, and 39% and 21% for adults as reported previously [26]). ARA-juv-5/ASV Ara-3 represents 12% of the reads for juvenile samples (and for adults 19% and 2%). The three other main adult ASVs (ASV Ara-2, -4, and -5) are present in juvenile gill bacterial communities in low abundances (below 1%). ASV Ara-2 represents 0.02% and 0.16% of reads for the two juveniles (in adult 32% and 13%); ASV Ara-4 represents 0.13% for the first individual and is absent in the other (in adult 1% et 13%); and finally, ASV Ara-5 represents 1.01% and 3.27% of reads (for adults 10% and 2%). In the tree, ARA-juv-7 appears as the closest relative, yet not identical, to two adult ASVs, namely, ASV Ara-2 and -4 (Figure 6).

*Minuca rapax* juveniles shared 13 ASVs with burrow water samples (Figure 4). The four ASVs that are abundant in *M. rapax* juveniles are all found in burrow water samples. The juvenile-dominant ASV MIN-juv-1 represents 10% of the reads for burrow waters, MIN-juv-2 represents 0.8%, MIN-juv-3 represents 0.3%, and MIN-juv-4 represents 0.2% of the reads (averaged from the two burrow samples). The other ASVs shared between *M. rapax* and burrow waters represent between 0.05% and 2.2% of the reads for *M. rapax* and 0.03% and 7.23% for burrow water samples. The ASVs shared were present only in one burrow water sample.

A comparison was then made then with the five main ASVs previously found in adult *M. rapax* gills, namely, ASV-Min-1, -2, -3, -4, and -5 [26]. One ASV (MIN-juv-4) was 99% similar to ASV-Min-2 found in adult specimens, (5 nt difference over 350 nt) and appeared as the closest relative in the tree (Figure 6). MIN-juv-4/ASV-Min-2 represents 1% of the reads for juvenile samples, and 2% and 20% for adults [26].

Two other ASVs, each rare in *M. rapax* juveniles, were 100% identical to adult ASV-Min-4 and ASV-Min-2:

ASV-Min-2 is identical to one rare ASV found in juveniles that represents 0.09% of the reads. ASV-Min-4 is identical to one rare ASV in juveniles that represents 0.08% of the reads. These rare ASVs were not found in burrow water samples. ASV-Min-1 and ASV-Min-5 were not found in juveniles.

## 4. Discussion

Crabs are very important in mangroves and are considered as ecosystem engineers due to their bioturbation action [44]. They consume litter and enhance the availability of organic matter, they recycle the nutrients of the mangrove, they aerate the sediment through their burrowing activity, and reduce soil salinity, which has an impact on mangrove growth [45]. The bacteria associated with the crab’s carapace also allow the bioavailability of nitrogen in the environment [46]. One study shows that there are symbionts associated with crabs [26]. The association between bacterial ectosymbionts colonizing soft tissue was also described in other well-studied invertebrates such as mussels colonizing sunken wood [47]. To understand the importance of symbiosis in this ecological role, we need to understand how individuals acquire it.

### 4.1. Gill Epibionts Are Environmentally Acquired

Vertical transmission of bacterial symbionts implies the presence of symbionts in the gonads and/or gametes, allowing transfer into embryos and larvae as well, as previously described in the insect *Blattela germanica* [48] or the bivalve *Solemya reidi* [49,50]. In *Aratus pisonii* and *Minuca rapax*, symbionts were detectable neither in the gonads nor gametes and larvae using PCR with three distinct primer sets and SEM on the eggs of *A. pisonii*. These data support that bacteria are either absent or present at undetectably low densities at these stages, and thus likely not vertically transmitted.

In the hydrothermal vent shrimp *Rimicaris exoculata*, the surface of eggs is colonized by bacteria [51], but the authors suggested a horizontal transmission mode, with these bacteria being acquired from the environment rather than parents. Later in development, bacterial symbionts are absent from the gill cavity of shrimp larvae, supporting that the symbiosis-competent bacteria come from the environment [52], in which free-living very close relatives of the major gill-colonizing Epsilonproteobacteria (Campylobacterota) are documented to colonize available substrates [53].

In this present study, gill bacteria were detected in the juveniles of mangrove crabs; thus, the transmission mode seems to be environmental. Among potential source habitats, the surface of *Rhizophora mangle* did not share any ASVs with either crab species. For some organisms whose mode of transmission is environmental, it sometimes happens that some associated symbionts are not found in the living environment [54]. However, two of the main ASVs found in *A. pisonii* juveniles (ARA-juv-1, ARA-juv-2) are highly similar to ASVs found in burrow waters. At this level of similarity, we consider them to belong to the same bacterial species. ARA-juv-1 is the main ASV colonizing the juveniles. This implies that the main ASV in *A. pisonii* is found in the burrow water of *Minuca rapax*, suggesting burrow water could be a source of symbiosis-competent bacteria. In *M. rapax,* all four abundant ASVs present in juveniles are also present in their burrow waters. It is thus likely that *M. rapax* juveniles acquire their symbiotic community from the environment. We assume that the acquisition of symbionts in both species occurs after the first molt of megalopa into a young juvenile crab.

### 4.2. Community Similarity between Juveniles and Adults as Evidence for Host–Symbiont Fidelity

In a recent study, the main ASVs occurring on the gills of *A. pisonii* and *M. rapax* were characterized [26]. The ASVs of juveniles (what we call here “the community”) were compared to the main ASVs of adults of the same species. In the present work, all of the main ASVs from adults of *A. pisonii* were found in the juveniles of the same species. Two ASVs that are predominant in adults are also dominant in juveniles; the others are otherwise rare in juveniles. The most abundant ASV in adults is also the most abundant in juveniles. For *M. rapax*, three of the five major adult ASVs are present in juveniles; of these three ASVs, one is abundant in juveniles, while others are present in low proportions. However, the most abundant ASV in juveniles (MIN-juv-1) is absent among adult-dominant ASVs, and it could be a life-stage-specific symbiont. The conservation of major ASVs from juveniles to adults is intriguing given that the gills of post-molt adults of *A. pisonii* appear devoid of symbiont. In *Rimicaris exoculata,* bacterial symbionts attached into the gill chamber are also eliminated after each molt and the symbiotic community is re-established in the gill chamber gradually [55]. A similar process was observed for the lobster *Homarus gammarus*, whose gills are re-colonized by filamentous, cocci, and rod-shaped bacteria after each molt [21]. Thus, observation of post-molt *A. pisonii* gills devoid of bacteria suggests a similar recolonization occurring in mangrove crabs. The fact that similar bacteria are found in juveniles and adults despite renewal after molt suggests a level of host–symbiont fidelity through time, as observed in other crustaceans for which post-molt recolonization involves the same bacteria that were present pre-molt [55].

From the juvenile stages on, the bacterial community composition tends towards that observed in adults. This means the bacterial community is not yet “adult-like” at the first juvenile stages we observed in this study for theses crabs. This is similar for the shrimp *R. exoculata,* in which the bacterial community switches throughout embryonic development and is somewhat influenced by the chemistry of the environment [51,52].

The juveniles of *Aratus pisonii* develop in a different biotope compared to the adults. Indeed, *A. pisonii* juveniles share their biotope with juveniles and adults of *Minuca rapax*. Despite this common biotope and the fact that some of the juvenile ASVs (or very closely related ASVs) of both species occurred in burrow waters, juveniles of *A. pisonii* and *M. rapax* display different associated bacteria. This is not surprising, as even if the environment can have an effect on the community in certain cases [56], the host species is usually a stronger driver of the composition of the bacterial community. This was previously observed in the gut microbial community of freshwater shrimps, where less OTUs are shared when the host species is different. Zhang et al. (2016) [24] also demonstrated that the constitution of the community depends on the host’s genetic factors as well as on the different tissue localizations rather than on the environment, indicating that an (unknown) selection mechanism occurs.

The role of this association is still unknown. It could serve to feed the host, as is the case in other organisms whose symbionts are located on the gills [57]. Symbionts could also play a protective role, as in the case of the pea aphid *Acyrthosiphon pisum* and its symbiont *Hamiltonella defensa* [58]. Indeed, a recent study found antimicrobial molecules associated with gills and bacteria of *A. pisonii* and *M. rapax*. This new information reinforces the idea of a protective role for this association [59].

## 5. Conclusions

The absence of symbionts in gonads, gametes, and larvae suggests an environmental acquisition of symbionts in both *Aratus pisonii* and *Minuca rapax*. This hypothesis is supported by the presence of bacteria in juveniles, some of which also occur in burrow water. The colonization of crab gills by bacteria must occur after the megalopa stage. Indeed, depending on the species, gills can be present from the first larval stages (zoea 1); however, they are often present as undifferentiated gill buds. As development proceeds, the gills become fully developed and functional in the first crab stage obtained after the megalopa 60–62]. Symbiont acquisition is thus probably dependent on the presence of functional gill filaments in juveniles. We hypothesize that the gills must be completely developed to acquire the symbionts, as observed in the bivalve family Lucinidae [14,15].

In the future, monitoring gill community compositions throughout hosts’ lives will help to better understand their establishment. For this, being able to follow the development of mangrove crabs after their larval phase in the laboratory would be a highly valuable tool in order to test which environmental sources of bacteria are used, identify the mechanisms of symbiont selection, and decipher the role of gill bacterial symbionts.

## Figures and Tables

**Figure 1 microorganisms-12-00652-f001:**
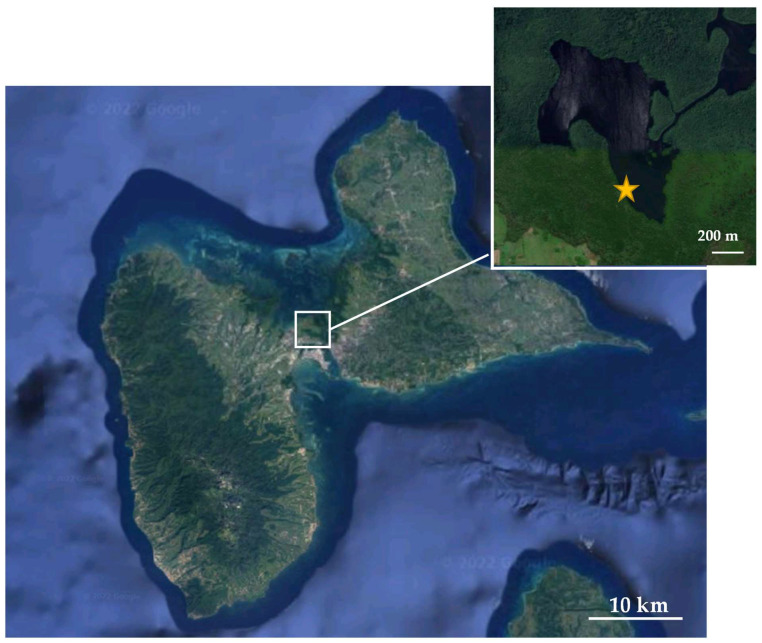
Archipelago of Guadeloupe, Lesser Antilles. Site of collection: Manche à Eau (white square, yellow star).

**Figure 2 microorganisms-12-00652-f002:**
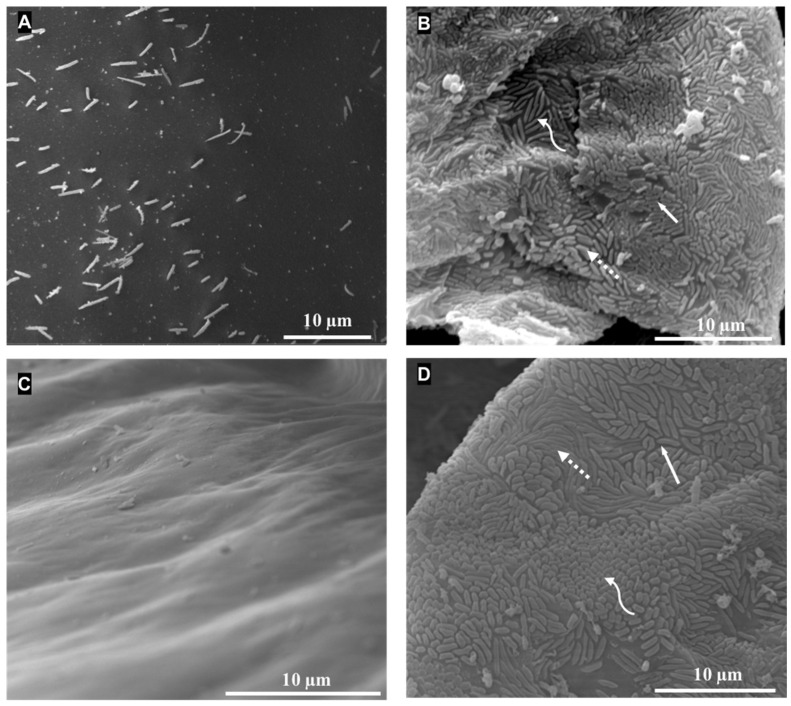
SEM pictures of eggs and gills after molt (adult) of *A. pisonii*. Pictures of gills of *A. pisonii* and *M. rapax* juveniles. (**A**) Egg surface of *A. pisonii* is devoid of bacteria while presenting several outer cell ornamentations. (**B**) Gill surface *A. pisonii* juveniles covered by three different morphotypes of rod-shaped bacteria—white arrow: short and thin morphotype; white dotted arrow: short and big morphotype; white curved arrow: long morphotype. (**C**) Gill surface of *A. pisonii* individual captured in the field just after molt, no bacteria were seen on the gill lamellae. (**D**) Gill surface of *M. rapax* juveniles covered by three different morphotypes of rod-shaped bacteria—white arrow: big morphotype; dotted arrow: thin morphotype; curved arrow: short morphotype.

**Figure 3 microorganisms-12-00652-f003:**
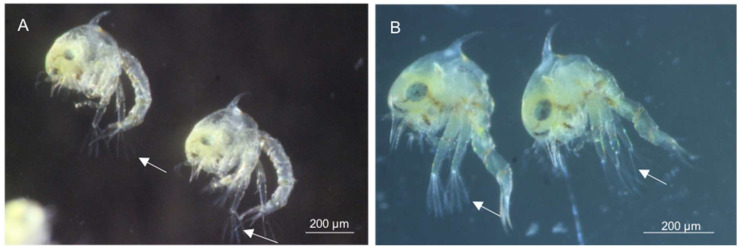
The larvae of *M. rapax* and *A. pisonii* species. First larval stage obtained laboratory culture: (**A**) *Minuca rapax* zoea and (**B**) *Aratus pisonii* zoea. The eye, the dorsal spin, and the rostral spin are visible on the carapace of the zoea. Additionally, 4 plumose marginal setae are visible at the end of the exopods (white arrow).

**Figure 4 microorganisms-12-00652-f004:**
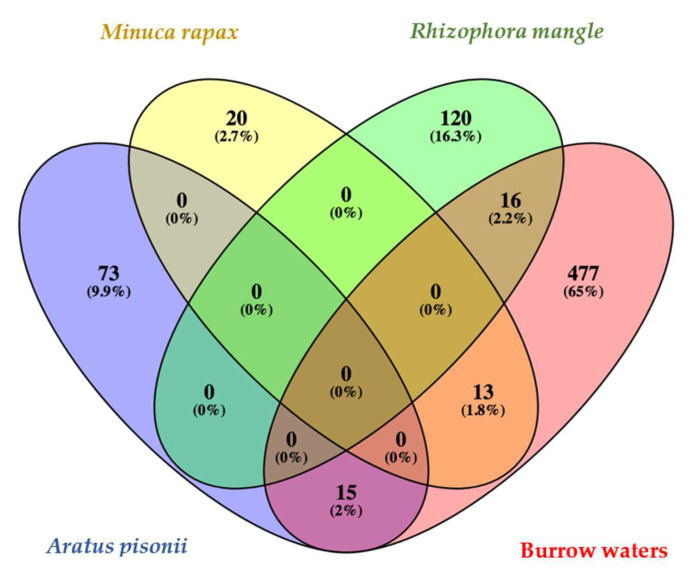
Venn diagram between *A. pisonii* juvenile gill, *M. rapax* juvenile gill, burrow waters, and *Rhizophora mangle* branch sample. In total, 88 ASVs are found on juvenile gills of *A. pisonii* (both samples combined), 136 ASVs are found on Rhizophora mangle surface (both samples combined), 33 ASVs are found on *M. rapax* juvenile, and 521 ASVs are found in burrow water samples.

**Figure 5 microorganisms-12-00652-f005:**
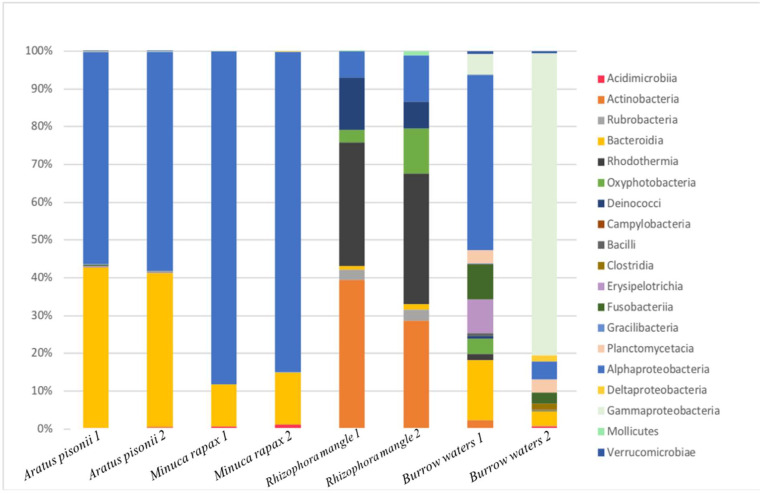
Bar plot showing the proportions of bacterial classes as percentages of reads in the different samples.

**Figure 6 microorganisms-12-00652-f006:**
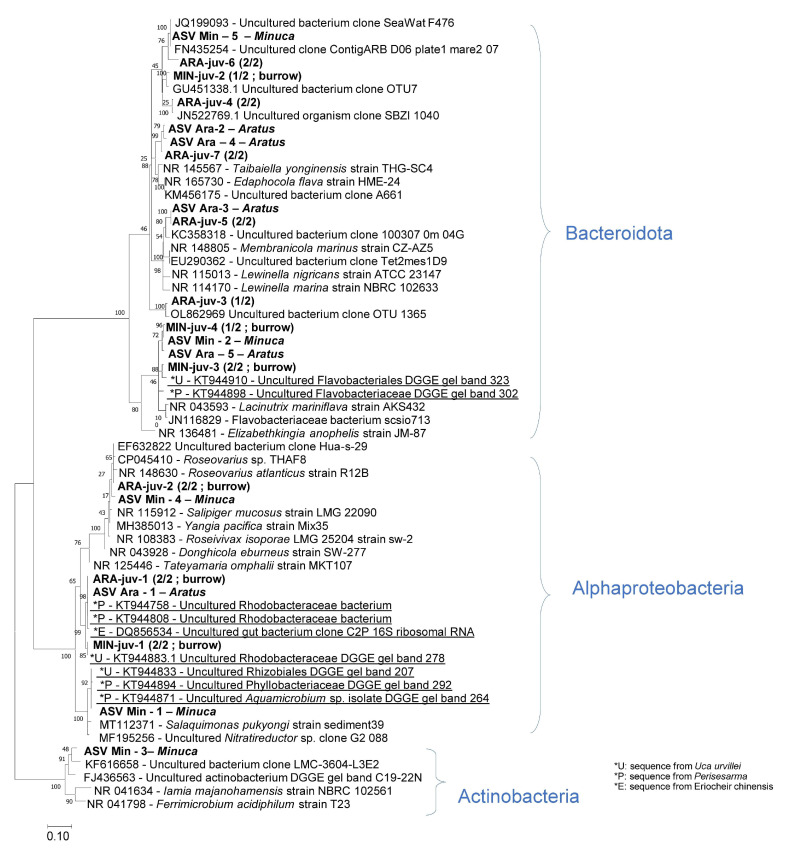
Phylogenetic relationships of the 7 and 4 abundant ASVs found in juveniles of *Aratus pisonii* and *Minuca rapax*, respectively labelled as ARA-juv-x and MIN-juv-x (in bold). In parentheses: number of specimens (out of 2) in which the sequence was recovered; “burrow” indicates that a highly similar to identical ASV was found in at least one burrow water sample (see text). The 10 dominant ASVs previously identified in adult *A*. *pisonii* and *M*. *rapax* [26] also appear in bold under names ASV-Ara-x and ASV Min-x, respectively. Sequences obtained from other crab genera (*Uca*, *Perisesarma* and *Eriocheir*) are underlined. The scale bar represents 10% estimated sequence divergence; percentages at nodes were calculated based on 100 bootstrap replicates.

**Table 1 microorganisms-12-00652-t001:** Summary of the different analyses performed on the study samples; “nt”: not tested; “+”: presence of bacteria for the sample tested; “-”: no evidence for the presence of bacteria.

	*Aratus pisonii*						*Minuca rapax*				Environmental Samples	
Techniques/Samples	Gonads	Eggs	Wild Larvae, Megalopa	Laboratory Larvae	Juvenile Gills	Molted Adult Specimen	Gonads	Eggs	Laboratory Larvae	Juvenile Gills	Burrow Waters	*Rhizophora mangle* Root’s Surface
Universal PCR (8F-1492R)	-	-	-	-	+	nt	-	-	-	+	+	+
PCR (8F-CFB 563R)	-	-	-	-	+	nt	-	-	-	+	+	+
PCR (ROS 537 R-8F)	-	-	nt	-	+	nt	-	-	-	+	+	nt
Scanning Electron Microscope	nt	-	nt	nt	+	-	nt	nt	nt	+	nt	nt
16s rRNA sequencing (V4/V5 region)	nt	nt	nt	nt	+	nt	nt	nt	nt	+	+	+

## Data Availability

All data are available on GenBank NCBI.

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
