# Peer review of "Environmental Transmission of Symbionts in the Mangrove Crabs *Aratus pisonii* and *Minuca rapax*: Acquisition of the Bacterial Community through Larval Development to Juvenile Stage"

_microorganisms, 2024, doi:10.3390/microorganisms12040652_

Round 1

Reviewer 1 Report

The manuscript aims to investigate the transmission of bacterial symbionts in the different life stages of two species of mangrove crabs (Aratus pisonii and Minuca rapax). More specifically, the authors aim to infer whether vertical or horizontal transmission is the main mode of symbiont transmission among this crab species by investigating the presence of microbes at the different life stage of the hosts (gonads/eggs to adults) and in the surrounding environment using molecular approach (PCR) and imagery (electron microscopy). Using these approaches, the authors only observed the presence of bacteria on juvenile gills suggesting a gradual establishment of the symbionts during the life of the crabs starting from the megalopa stage and involving epibiosis-competent bacteria that occur in the environment.

Overall, the manuscript is relatively well written and its aims on investigating the mode of transmission of symbionts among crustaceans is totally of interest and this study has the potential to help flesh out that research area. However, there is too many gaps and/or errors in the methodology used and presented in this study, that I cannot recommend the publication of this version of the manuscript in Microorganisms. The manuscript should be subject to a major revision before I can consider the validity of these results and judge if their article could be potentially accepted or rejected for publication.

Having received a document without line numbers that I can use to refer of some specific comments, I will here stick to list my major concerns that I hope the authors will be able to clarify and/or discuss. Furthermore, the authors mentioned in their manuscript some supplementary data that I didn’t have access too. I will appreciate, if a second round of revision will be performed, to have a complete version of the manuscript (include those supplementary data) and a document with line numbers a to facilitate my work as reviewers and 

My major concerns for the proposed study are the following ones:

1/ The actual numbers of samples/individuals collected and analyzed in this study and how the samples were prepared for DNA extraction. 

Apart if I missed it, nowhere it is said how many females and juveniles from both species were collected and how many environmental samples were taken at each sampling time point (any replicates?). Also, it is unclear how many samples from each species and life stages were analyzed by electron microscopy. Please clarify.

But more importantly, it is unclear why only two adults and two juveniles were only analyzed by molecular approaches. I think having only two replicates per life stages and species is insufficient to perform a proper comparison. This is in my eyes problematic in the aim of this paper. Looking at the number of ASVs found in each samples suggest that a non-negligeable part of the retrieved richness is sample-specific and few are shared between them. For instance, for the two Minuca rapax juveniles, 17 ASVs for both samples were found (so a total of 34 ASVs) but combined the richness in this species represent 32 ASVs, meaning that only two ASVs are shared between samples. This is difficult to then define the community of symbionts community from a given crab’s species and at a specific life stage when so few are shared between replicates. I believe the only way will be to analyze more samples. With such low shared diversity between replicates, my first impression was that the observed diversity is not composed by proper symbionts but by organism “accidentally” attached to the gills of the juveniles coming from the environment that are not meant to be their symbionts. I believe the recurrence of ASVs present in the juvenile’s gills and retrieve also in the surrounding environment is the only way to define their symbionts (and having more samples will “solve” that issue). How are confident are the authors that they describe only the symbionts community? How was “cleaned” the samples to remove any residual rDNA coming from the environment? And why have they limited their analysis to only two juveniles.

2/ The description and discussion of the obtained prokaryotic diversity. The authors have described the obtained diversity at either low taxonomic resolution (phylum level) or ASVs level but can they assign any of the ASVs to any class, genus or species level? It will be interesting to know more specifically the identity of the retrieve ASVs to see if this ASVs could indeed be symbiont of those crustaceans. A discussion of their potential role as symbionts will have be interesting to have too. 

3/ The metabarcoding analysis of juvenile’s gills, environmental samples and adults coming from two different studies. and the comparison of the two crab’s species coming from two different sampling time point. 

As outcome of this study, if I am understanding correctly, is that the establishment of symbionts in the gills of the studied crabs starts at the juvenile stage and is transmitted to the adult but has for basal diversity organisms coming from the surrounding environment. Meaning that the first mode of transmission will be horizontal (environment to juvenile) then vertical (juvenile to adult). Am I correct? If so, the first established community of symbionts in juveniles, and subsequentially adults, will be factor of the environment and period of sampling and consequently, samples/ individuals taken at different season/location will give different results and this is what the authors see. But in order to show that indeed the symbionts are coming from the environmental and are not host specific, it will have made sense to study juveniles and adults of both species coming from the same sampling period AND from a different time point. In this manuscript, we have only one side of the aspects, where adults and juveniles are coming from different study (so I guess different sampling time) and where the two species of crabs were sampled at different period too. In this manuscript, it misses the results that confirms indeed the environment is the only ‘source’ of symbiont and this could be obtained when both species are sampled together. As their juveniles are living in the same habitats, the symbionts communities should be very similar. 

Furthermore, it is unclear how the authors did the comparison of the diversity analysis by DADA2 from the two sequencing datasets (from this study and Béziat et al., 2021) to show the similarity in the prokaryotic community between juveniles and adults. Did they perform a sequence alignment? if so, how? Details have to be given to know how the ASVs comparison from both studies were made and a figure/table to show that comparison is needed. 

Yet, I believe that the proper way to highlight the vertical distribution of symbionts between juveniles and adults required that samples are taken at the same time. Since in this manuscript, this is not the case, I will expect some justification and discussion on that aspects.

4/ The use of PCR before metabarcoding (Miseq) to detect the presence of symbionts. As I understand, the authors performed PCR using several sets of primers in the different life stage of the two species of crabs (from gonads to juvenile’s gills) before performing metabarcoding on the samples that only gave originally positive results with the PCRs. This in order to save time and money on focusing only on samples that have confirmed the presence of symbionts, which is understandable. Is that correct? If so, this need to be said clearly to understand the reason of using both approaches. Did the authors have envisioned to send one of these DNA samples with negative results at the PCR (for each life stage and species) to confirm the first observation made with PCR? 

5/ Please provide the accession numbers of the sequencing datasets of this study.

Reviewer 2 Report

On manuscript on Schanendra et al. on the role on environment on gill bacteria symbionts on two crab species (Aratus pisonii and Minuca rapax). Although manuscript on relevant data on lacks some clarifications. On began on the impacts on these two crab species? I mean why on these chosen on study? On what impacts on the environment? Another remark refers on the lack on previous investigations on these two crab species? On where they studied elsewhere? On this a first report? Other remarks refer on authors conclusions. Authors state that environment bacteria on contribute on gill community and that this may on be found on adults. I mean on symbionts on this a unique relation on crabs on comparison on other organisms? I mean what on authors first hypotheses on the symbiotic community on these two crab species?

Finally, on ecological role on this environmental-driven symbiotic gill bacteria community on these two crab species on mangroves?

Minor editing of English language required

Reviewer 3 Report

This manuscript describes an investigation into the mode of transmission of symbiotic bacteria in two species of crabs living in different mangrove habitats. The results, obtained through PCR and microscopic observations, indicate that the transmission is horizontal, with the acquisition of bacteria from the environment at the juvenile stage. It was also interesting that symbiotic bacteria differ between the two species, despite their living in the same area. These findings represent a significant contribution to the field.

 However, the presentation of the figures, tables, and text requires improvement, as their correspondence is not clear. Several points require further clarification for the manuscript to be accepted.

 The inclusion of photographs of the organisms and tissues (adults, juveniles, larvae, eggs, and gonads) is recommended. Additionally, it would be helpful to provide information regarding the gill collection method.

 Table 1 showed that no amplification occurred using universal primers for larvae, which is surprising because presence of at least some gut microbes is expected. I afraid that the initial sample amount was insufficient to extract DNA. So, authors should test universal primers for 18S rRNA as a positive control.

 To enhance the clarity of Figure 2, it is suggested that pictures with the same magnification in each tissue be included.

 A table for genus level is needed to supplement the graphs of the phylum level. This table should include the previous results of the adult samples, at least for the major ASVs, to facilitate following the text.

Round 2

Reviewer 1 Report

The authors have done a thorough work with the revised version and addressed and/or defended many of the issues that I have raised. Their manuscript has greatly gained in clarity and I am overall happy with the new version of the manuscript.

However, before making any final decision and endorsing the manuscript, I still have minor comments/remarks to address in order to clarify (and eventually improve) some aspects of the manuscript.

L15-16: This statement made at the beginning of the abstract is referring to the previous work of the authors done on adults of these two species of crabs, am I right? Please clarify in order to avoid any confusion with what the authors bring in this new manuscript. Otherwise, it looks that this is something already knew and we don’t see the relevance of the present work.

L22: Replace “PCR test” by “PCR tests” as several primer sets were used.

L28: Could “on the other hand” be replaced by “however”? For me (but I might be wrong), “On the other hand” is usually used for expressing an opposition of though/statement and in the context of this abstract I don’t see such opposition.

L40: “living together”” instead of “Living together”, no?

L57: Should it be “Wolbachia spp.” instead of “Wolbachia”?

L62-63: Could the authors insert here any references to support this statement showing that symbioses have been more documented in Mollusks?

L67: Maybe replace “suggesting environmental transmission” by “suggesting environmental transmission for those species”? This, in order to clearly said that, apart for the case of the Kiwa crabs, there is no study (or few) that prove the environmental origin of crab symbionts and then highlight the “need of the present study”. 

L72-75: Here, the authors introduce for the first time their previous work conducted on adults of the studied crab species. I wonder if the authors have also to clearly say on this part of the introduction that the proposed study is the follow-up of this previous work by looking at the other stages of the life stage of these crab species. This, in order to “justify” the comparison between the two studies and to induce any reader to also consider this previous work to understand the present work.

L72-75: This is an information that I think could be relevant to know about those two species of crabs in the context of the symbiont colonization and transmission, which concerns their life stages.  Do the authors have any idea of how long each life stage of these crab species last? Are they the same between species? This, in order to give an idea to the reader of which time scale of colonization and transmission we are talking here (days?  months? ...).

L84-85: Can the authors add some references relative to the habitats of the adults and juveniles of the two species?

L117-118: More details have to be given on how water samples were collected. As I understand the authors took 4 water samples using a 50 ml syringe that they pooled prior to DNA extraction but how much volume did they take each time (50 ml? if so, it is a total of 200 ml?) and how the pooling was done? I guess the authors have filtered the water samples on filter (as it is commonly done for such sample). If so, have they done some prefiltration? Which type of filter were used for collection (polycarbonate filters?)? Which diameter had the filter? which pore-size to collect all the prokaryotic community (0.2 um?)? Please clarify. 

Furthermore, I suggest to rephrase this sentence and simplify it as follow (if I am correct):

“XXX ml of water samples from several burrows of M. rapax were collected in duplicates at low tide using a 50 ml-syringe (pool of 4 independent samples)”. Here replace “XXX” by the actual taken volume.

L120-121: The authors said that “several roots and branches were scraped to make one sample”. Can the authors specify how much material (in gram?) this scrapping represent?

Furthermore, replace “Two were used for DNA analysis” by “DNA of two out these four samples were further extracted”. I overall think that the term “sample were used for” could be directly replace by a verb describing the action (for instance, “sample were used for extraction” could be replace by ““sample were extracted”). Please correct throughout the manuscript.

L137-140: Replace “crude DNA” by “Environmental DNA” to clearly identify that by crude DNA the authors refer to the ones coming from the surrounding environment. 

For the burrow water samples, I have some difficulty to understand how the authors actually proceed.  So as I understood they only took a total of 20 ml from 4 burrow samples, am I correct? Was it impossible to collect more? This is a limitation that the authors have to be aware and discuss in their paper I think (same remarks could be done for the scraped surface of R. mangle if the quantities of material were also small). 

Also, as I understood, the authors have then not filtered the burrow water samples prior to DNA extraction but have centrifuged it to collect a pellet that they have extracted. To my knowledge, this is something that are usually done for recovery DNA from algae or bacteria cultures (with very high cell density) but not (or rarely) on environmental samples. This would explain the low richness the authors retrieved that the authors could discuss about.  

L141-144: As I understood, all the samples (water, the scraped surface of R. mangle, life stage of crabs) were extracted using the same kits that are adapted to the DNA extraction of blood and tissue. Am I right? If so, do the authors really think that this kit is adapted to extract DNA of environmental samples? This need to be justified.

L145: Please cite the universal primers of eukaryotes used here for control. Please also specify the PCR condition (or cite a reference where these conditions can be retrieved).

Also, here and elsewhere (e.g., L147) for the different universal and specific primers of prokaryotes and bacterial groups, please give the reference of the original paper in which each set were designed and validated. One option could be to build a supplementary table with column containing the primers set name, their sequences, and reference that the authors will refer to here.

L145-147: Replace by “First, we performed a first screen for the presence of bacteria in various samples by PCR using universal primer set (8F-1492R).”

L147: Replace “Primer sets” by “Specific primer sets”.

L152-154: The authors used the same PCR conditions for all the primers sets tested here, right? Do all those set of primers have the same Tm? This is highly improbable. Please clarify.

L173-175: Please add reference to the different program/software and method used here (e.g. BLAST, CLUSTALW, Mega11).

L176: Please specify how the tree were built (which program?).

L189-190: This is not needed and can be deleted. Instead, the author should refer to this table when they mentioned results that are summarized here (for instance L213-214, L214-215, 230-232...).

Table 1: Maybe replace "Techniques/Samples" by "Techniques\Samples" and please specify that “16S rRNA sequencing” refer to the “16S rRNA (“V4/V5 region) amplicon sequencing” or (as not the entire 16S is sequenced here, this could be confusing).

L194: Replace “PCR amplifications from” by “Universal and specific PCR amplifications of the 16S rDNA gene from “.

L216: Please delete “(with any of the three primer sets used).”, which is not needed here.

L254/ Figure 4: These comparisons are cited later in the main text (after the figures 5 and 6), please replace the figures in the order they appear in the text.

L272:274: With the new figure 6 made by the authors, the presence of such table is now obsolete, the figure 6 shows already those results. 

L282/Figure 5: Please explain why only some % are shown on the barplot.

L291 and L292: Please refer to the supplementary table where these results come from.

L335/Figure 6: This is a great input but I believe this figure will be even better if the authors can add symbols and or colors to each of the ASV to refer to the sample they were found. Readers will be able to visually identify if the ASV is sample specific or can be retrieve in their surrounding environments or adult stage.

L594: Can the authors explain in the legend of this table what means "//" and what the two numbers (for instance 316/352 for the second cell) mean?

Furthermore, could the authors keep the same format for each of their table? These ones are 

the quality of the english is overall fine.

Reviewer 2 Report

Authors on adressed on comments and now manuscript warrants on publication.

Author Response

There was no new correction for this reviewer.